

# Development of multi field rock resistivity test system for THMC

Jianwei Ren[1], Lei Song[1,2], Qirui Wang[3,4], Haipeng Li[1], Junqi Fan[3], Jianhua Yue[1], Honglei Shen[1]

[1]State Key Laboratory for Geomechanics and Deep Underground Engineering, China University of Mining and Technology, Xuzhou 221116, China
[2]YunLong Lake Laboratory of Deep Underground Science and Engineering, Xuzhou 221116, China
[3]Institute of Defense Engineering, AMS, PLA, Luoyang, 471023, China
[4]Department of Civil Engineering, Tsinghua University, Beijing, 100084, China

*Correspondence to*: Lei Song (leisong@cumt.edu.cn), Qirui Wang (lywqr3061@163.com)

**Abstract.** In order to study the relationship between rock mechanical properties and resistivity under deep underground
environmental conditions, a rock resistivity test system was developed which can realize simultaneous control of high and low temperature, pressure, seepage and chemical environment, and the corresponding specimen sealing method was explored. The system mainly includes triaxial system, chemical permeation system, temperature control system and test control system. The reliability of the system was verified by tests and preliminary experiments. This device provides an important means to study the mechanical properties and resistivity characteristics of rocks in complex environments in deep-underground.

## 1 Introduction

There is a close relationship between rock mechanical properties and resistivity. Reflecting the changes of rock mechanical properties through resistivity is of great value in practical engineering (Yaramanci, 2000; Sun et al., 2015). It is an important method to study the relationship between rock mechanical parameters and resistivity through laboratory tests. Wang et al (2012) obtained the regression equation between resistivity and elastic modulus of rocks in various damage states by uniaxial
compression tests. yin et al (2021) established the relationship between damage variables and resistivity of sandstone by cyclic uniaxial compression tests; Kahraman (2021) studied the relationship between tensile strength, compressive strength and resistivity of volcanic clastic rocks by statistics of experimental results. The relationship between the mechanical properties of rocks and resistivity anisotropy during loading has also been studied (Chen et al., 2003; Jia et al., 2020). These studies have given a positive impetus to the use of resistivity for engineering rock monitoring.

As the underground development goes deeper, the environment of engineering rocks become more complex. Deep underground rocks are often subject to high temperature, high ground pressure, high seepage pressure and mineralized water corrosion, and even undergo freeze-thaw action in artificial freezing projects (such as liquid nitrogen fracturing, freezing method plugging), these factors will affect the mechanical properties and resistivity of rock (Ye et al. , 2015; Zhang et al., 2015; Ma et al., 2020; Gong et al., 2021; Tao et al., 2022). Therefore, it is necessary to study the relationship between rock
mechanical properties and resistivity under multi-field action. At present, a variety of devices are available for the study of rock mechanical properties in THM environments. Ranjith and Perera (2011) developed a high-temperature and high-pressure



triaxial test system for $CO_2$ storage research; Zhao et al (2012) developed a high-temperature and high-pressure triaxial system with the confining pressure reached 250MPa, the axial pressure reached 20MN, and the temperature reached 600℃; Huang et al (2020) designed a thermo-hydro-mechanical coupled triaxial device compatible with X-ray CT, with confining pressure,

axial force, seepage pressure and temperature up to 20MPa, 400kN, 10MPa and 100°C, respectively; Bai et al (2021) designed a triaxial cell with low-temperature loading function with a confining pressure up to 30MPa, an axial force up to 500kN, and a minimum temperature of -30℃. The application of these devices has greatly promoted the research on the mechanical properties of deep rocks, but they still do not have the function of testing rock resistivity under multi-field conditions.

In order to meet different research needs, researchers have developed a variety of triaxial devices with resistivity testing

functions. Alemu et al (2013) developed a resistivity test system that can control confining pressure and seepage based on the background of $CO_2$ flooding, which also has the function of real-time X-ray CT scanning; Bosch et al (2016) used resistivity tomography (ERT) as a resistivity test method to test the rock resistivity under confining pressure, axial pressure and seepage. But these two devices do not have the function of temperature control. Zhong et al (2010) designed a combined test system for rock wave velocity and resistivity that can control onfining pressure, temperature and seepage based on a conventional rock

triaxial instrument. However, the control mode and range of seepage pressure are not given in the paper, so it is impossible to judge whether it can meet the requirements of deep environmental conditions simulation. Falcon-Suarez et al (2014) developed a high-temperature triaxial seepage system with resistivity test function considering the seepage of brine-$CO_2$ in carbon storage, its maximum axial pressure is only 64mpa, which cannot meet the triaxial loading demand of dense rock. These devices are all based on the high temperature environment and use heating belts to control the temperature, which have high heating

efficiency, but does not have the functions of rapid cooling and sub-zero temperature control.

This paper introduces a new multi-field rock resistivity test system, which has the functions of rock triaxial test and resistivity test under the condition of high and low temperature, high pressure and high salinity water seepage. Its temperature control range is - 40 ℃ - 80 ℃, and the maximum confining pressure, seepage pressure and axial pressure are 30MPa, 20MPa and 400kN respectively. The device is suitable for studying the relationship between rock mechanical properties and resistivity in

complex environments.

## 2 Experimental system

The THMC multi-field rock resistivity test system is used to simulate the deep underground thermal–hydrological–chemical environment, and obtain the resistivity of specimen during loading. There are several difficulties need to be solved in the development of this experimental system: (1) Cooperative fine control of multi physical quantities under high pressure. The

axial pressure, confining pressure and seepage pressure are required to reach high loading capacity (axial pressure up to 400kN, confining pressure up to 30MPa, seepage pressure up to 20MPa) and have a wide temperature regulation range (-40℃ - 80℃). The above conditions need to be coordinated and fine controlled and can remain stable for a long time. (2) Real-time acquisition of multi-source information. It is also necessary to obtain the change of the resistivity of the sample in addition to collecting the conventional information of axial pressure, confining pressure, pore pressure, axial deformation, and hoop deformation.



(3) Anti-corrosion measures are required to ensure the durability of parts in contact with chemical solutions. And there should be enough insulation resistance between the resistivity test line and other parts to ensure the test accuracy.

The technical scheme shown in Figure 1 is adopted in this paper to solve the above difficulties. The test system includes triaxial system, chemical penetration system, temperature control system and test control system.

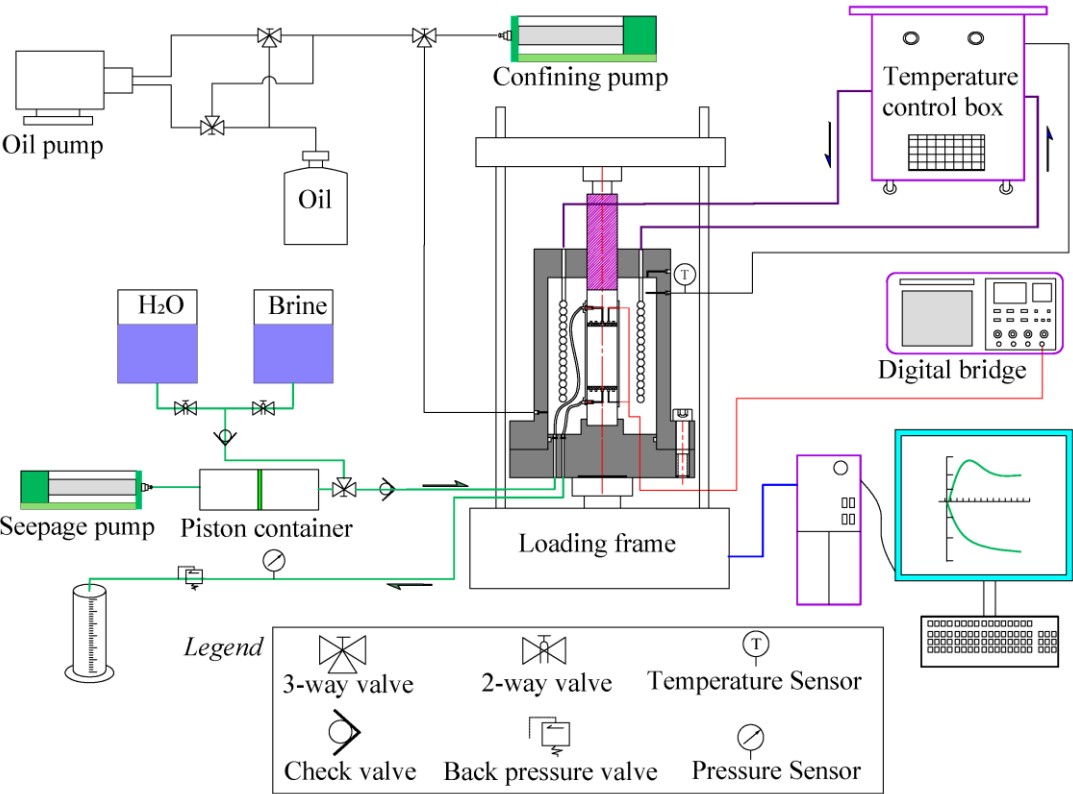

**Figure 1 Schematic diagram of THMC multi-field rock resistivity test system**

## 2.1 Triaxial system

The triaxial system consists of triaxial pressure cell, loading frame, confining pressure pump and oil filling device.

Triaxial pressure cell is the core component of the system which needs to organically integrate the functions of temperature control, resistivity test and chemical penetration, so as to meet the requirements of stress loading, high and low temperature

control, chemical solution seepage, and resistivity testing of the sample at the same time. The structure of the triaxial pressure cell was designed as shown in Figure 2. Triaxial pressure cell is mainly composed of cylinder, base, plunger rod, upper pressure cap, lower pressure cap, temperature control tube and seepage tube. The cylinder and the base are connected through flanges, which plays the role of overall enclosure and providing reaction force, and can withstand 400kN axial pressure, 30MPa confining pressure and 100℃ temperature. Considering the existence of chemical solution in the test, the cylinder and base

are made of rust-resistant 304 stainless steel. The plunger rod is located in the center of the top of the cylinder and transmits



the axial pressure provided by the reaction frame to the sample. The upper pressure cap and the lower pressure cap have the same structure and are composed of cylindrical insulating spacers and electrode plates. The insulating cushion block is made of high-strength PEEK material with seepage channel and wire interface. There are small holes on the electrode plate for seepage. The temperature control tube is made of spiral-shaped 304 stainless steel pipe, which controls the temperature of the

specimen by circulating the hot and cold medium inside. Considering the convenience of assembling and disassembling the sample, all pipeline and wire interfaces are arranged on the base. The permeation tube is made of stainless steel with high pressure bearing capacity. In order to avoid the short circuit of the test circuit caused by the contact between the seepage tube and the triaxial cell, a pair of PEEK material sealing plugs are set on the lower flange as the seepage tube channel. Non-conductive silicone oil was selected as the perimeter pressure loading medium due to the triaxial pressure cell has resistivity

test leads.

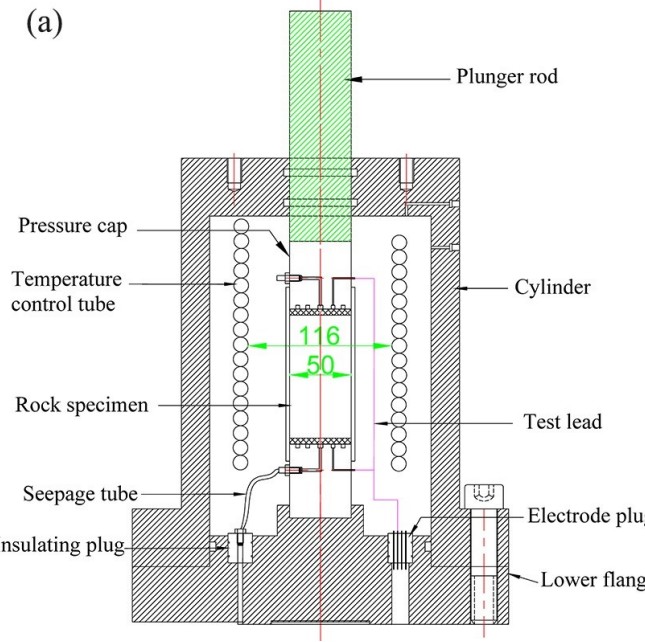
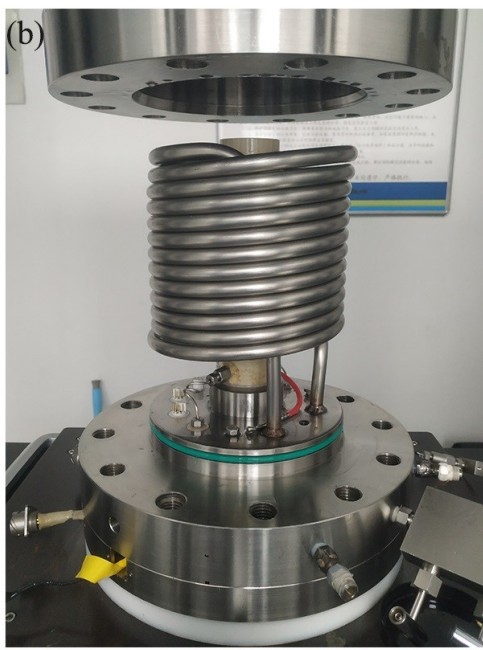

**Fig. 2 Triaxial pressure cell. (a) Design drawing. (b) Entity.**

The loading frame is used to provide axial load for the three-axis system, with 400kN loading capacity and 100mm loading range, and the speed control can be from 0.00001mm/min to 7.0mm/min. It can realize loading methods such as stress-

controlled loading, strain-controlled variable loading, and stress path loading.

The confining pressure is controlled by the constant current stabilized pressure pump which is pressurized by the piston driven by the stepping motor and adjusted by the feedback of the closed circuit. The volume change can be measured by calculating the number of steps of the stepping motor. Its pressure range is 0-32Mpa with an accuracy of 0.3MPa. The capacity and accuracy are 200ml and 0.01ml respectively.



The volume of triaxial pressure cell is about 6000ml, while the volume of confining pump is only 200ml. The efficiency of oil injection to triaxial pressure cell by confining pump is too low, so an individual oil filling device is designed. It is composed of an oil pump and pipes with valve, the oil pump is a one-way oil delivery device, which can realize the oil injection and drainage of triaxial pressure cell through valve adjustment. The filling device and confining pressure pump are connected with the triaxial pressure cell through the three-way valve. After the triaxial pressure cell is assembled, adjust the valve to connect

the oil filling device with the triaxial pressure cell, allowing oil to be injected into the triaxial pressure cell through the oil filling pump. When the triaxial pressure cell is full of oil, adjust the valve to connect the confining pressure pump with the triaxial pressure cell, and apply confining pressure to the specimen.

**2.2 Chemical seepage system**

The chemical seepage system is used to simulate the deep underground chemical environment. It mainly includes constant

current stabilized pump, piston container, back pressure valve, liquid storage tank. Constant flow stabilized pressure pump is to provide the required seepage pressure for the system, which has the same parameters as the confining pressure loading pump. The piston container is located between the constant current stabilized pump and the liquid storage tank, which separates the chemical solution from the pump body to avoid corrosion of the constant current stabilized pump by the chemical solution. The back pressure valve is installed on the side of the water outlet hole of the rock sample to maintain the stability of the pore

pressure.

The piston container is a hollow cylindrical container separated by a piston. When a liquid under pressure is introduced on one side, the pressure can be transferred to the liquid on the other side by pushing the piston. The piston container is made of 2205 stainless steel, which has a volume of 1000ml and a pressure resistance of 40MPa.

**2.3 Temperature control system**

The temperature control is regulated by the high and low temperature circulation box, which is connected with the temperature control tube in the triaxial pressure cell through the pipeline. Silicone oil is used as circulating medium to take into account high and low temperature control. The high and low temperature circulation box includes a refrigeration system and a heating system that can work independently or in coordination. And it can be feedback-adjusted according to the temperature in the triaxial pressure cell. The temperature adjustment range of the high and low temperature circulation box is -40~200°C with an

accuracy of 0.5°C.

**2.4 Acquisition control system**

The acquisition control system includes strain acquisition and resistivity acquisition, and can control the loading rack and constant-current stabilized pump through software. The axial strain and radial strain of the specimen are measured by the temperature self-compensated foil strain gauge. Two strain gauges are pasted on the middle of the specimen side, one of the

strain gauges is parallel to the axis of the specimen to measure the axial strain of the specimen, and the other is perpendicular



to the axis of the specimen to measure the radial strain of the specimen. The axial and radial deformation of the specimen can be displayed in real time through the acquisition software.

The resistivity measurements were performed following the two-phase electrode method shown in Figure 3. The overall resistance of the rock specimen is measured by a digital bridge, and the average resistivity of the rock specimen between the two electrodes is calculated according to Eq. (1):

$$\rho_0 = R\frac{S}{L}, \tag{1}$$

Where $\rho_0$ is the average resistivity of the specimen, $\Omega\cdot m$; $R$ is the resistance value of the test piece, $\Omega$; $S$ is the cross-sectional area of the test piece, $m^2$; $L$ is the length of the test piece between the two electrodes, m. In order to prevent the electrode from being polarized during the test, the electrode is made of thin copper plate and the test frequency is set to 100Hz according to Zhong's (2010) practice.

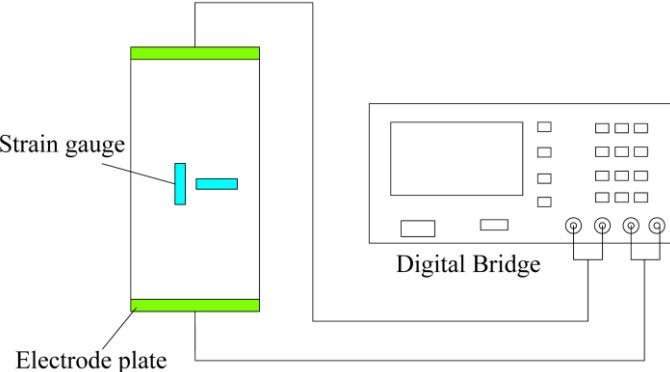

**Figure 3 Schematic diagram of resistivity test system.**

### 3 specimen sealing

When conducting rock resistivity measurement under triaxial conditions, a flexible protective film (heat shrinkable film or latex film) is often coated on the surface of the specimen to prevent the immersion of confining fluid from affecting the mechanical properties and resistance test of the specimen. However, the surface of the specimen is not completely smooth, and the protective film may not fit tightly with the sample. When the specimen is saturated or supplied with pore water, a through water film will be formed on the specimen surface, causing the current path to change. At this time, the measured resistance is the paralleling of the specimen resistance and the water film resistance, which may cause errors in resistivity measurement. In order to evaluate the influence of the water film on the resistivity measurement, the resistivity of specimen with different water film thickness was measured.

### 3.1 Water film impact test

The test steps were: first, the specimen was dried and weighed, then saturate the specimen with 20g/L of sodium sulfate solution and weighed again, the excess water on the surface of the saturated specimen was wiped off before weighed; finally,





155 the specimen was coated with conductive paste at both ends and connected to the test electrode and placed on the electronic

balance as a whole to test the weight and resistance of the rock at different evaporation times. The test system is shown in

Figure 4. The specimen used was limestone with a porosity of 0.66%, and the ambient temperature of the test was 26.8°C and

the humidity was 44.1%. Figure 5 shows the experimental results.

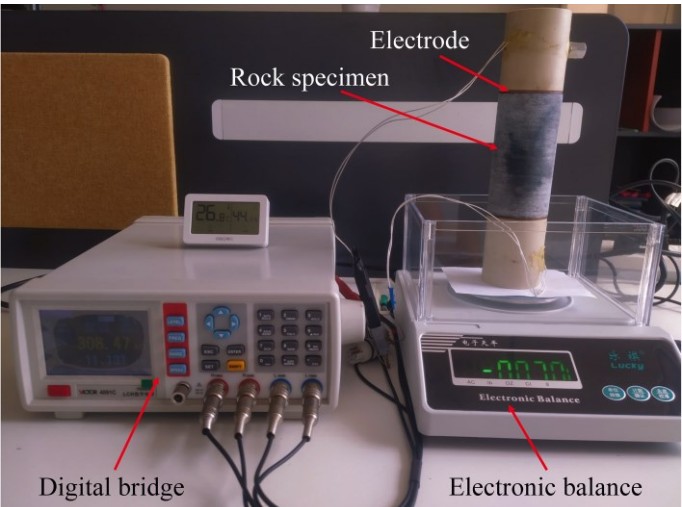

160 **Figure 4 Water film impact test system.**

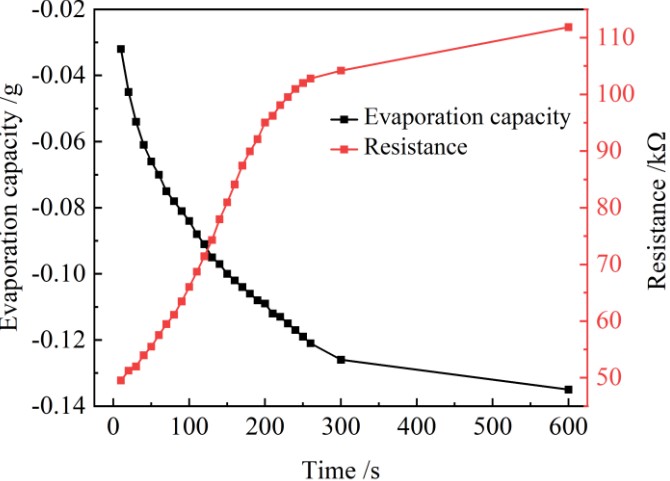

**Figure 5 Water film evaporation and resistance curves with time.**

It can be seen from Figure 5 that the rock weight gradually decreases with time, and the rate of decrease gradually decreases;

the rock resistance gradually increases, and the rate of increase shows a trend of first increasing and then decreasing. This is

165 mainly due to the large area and good connectivity of the water film on the surface of the specimen at the initial stage, which

makes it have good current conduction capacity; with the evaporation of water, water film area contraction, the current

conduction path changes sharply so that the specimen resistance changes more; finally, the water film is distributed in isolation,

 

and the influence of surface water film change on current conduction is reduced. Due to the short evaporation time, it can be considered that the reduction in 5 minutes are the water on the surface of the specimen, Within 5 minutes, the weight of the sample decreased by 0.12g, the resistance increased by one time, and the mass of water remaining in the specimen was 0.40g. It can be seen that although the surface water film accounted for a small proportion, but it plays a comparable role in the conductivity of the water inside the specimen, so measures need to be taken to avoid the formation of water film on the side of the specimen.

### 3.2 Specimen sealing method

To prevent the water film on the side of the specimen from affecting the resistivity measurement, a new specimen sealing method is proposed, the specific steps are: 1. Dry the sample at 105℃~110℃; 2. Apply a layer of liquid rubber plastic on the side surface of the sample with a thickness of 1~2mm; 3. When the rubber and plastic are about to solidify, cover the Teflon heat shrinkable tube outside the upper and lower cap and the specimen, and heat it with a hot air gun to make the heat shrinkable tube fully fit with the specimen surface; 4. Keep it for 24 hours to ensure that the rubber plastic adhesive is completely cured, and then saturate the sealed specimen. The sealed specimen is shown in Figure 6.

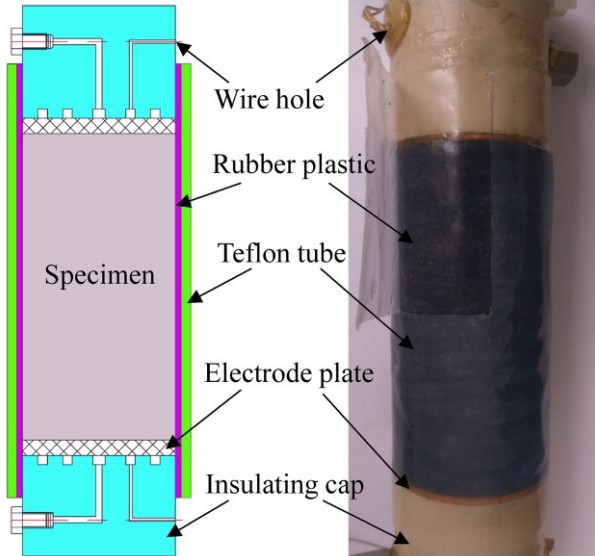

**Figure 6 Sealed specimen.**

### 4 System performance test

### 4.1 Sealability test

In order to test the effect of the new sealing method of the test piece, a comparison was made with two other sealing methods (heat shrink film sealing, latex film sealing method). The resistivity of the specimens sealed by the three methods was measured under different confining pressure conditions. Before the test, specimens were saturated with 20g/L $Na_2SO_4$ solution. The




result is shown in Figure 7. It can be seen that the resistivity of the specimen sealed by the new method increases linearly with the confining pressure, and the resistivity of the specimen sealed by the other two methods increases rapidly first and then

slowly with the confining pressure, and finally tends to be consistent with the resistivity of the specimen sealed by the new method. The resistivity of the samples sealed by different methods varies greatly under low confining pressure, which may be caused by the water film on the side of the sample. Therefore, the initial resistivity of saturated specimens sealed by different methods were compared with the resistivity of the evaporated specimen. It was found that the initial resistivity of the specimen sealed by the new method was close to that of the specimen after 10 min of evaporation, while the initial resistivity measured

by the other two sealing methods were lower than the initial resistivity of the evaporated specimen. This indicates that the new specimen sealing method can play a role in preventing the formation of a water film on the side of the specimen, which is more significant when the confining pressure is low.

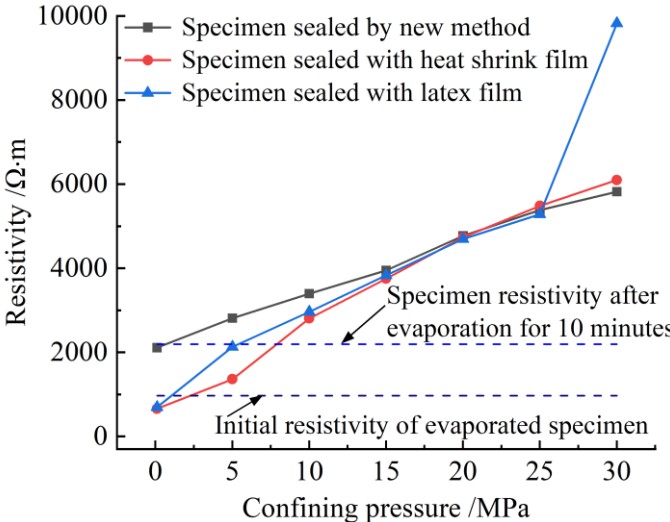

**Figure 7 Changes in resistivity of sealed specimens by different methods under confining pressure.**

**4.2 Resistance measurement accuracy test**

In order to avoid large additional impedances (such as contact resistance, wire resistance, etc.) in the resistivity measurement results, the variable length method was used to verify the accuracy of the measurement. The method is to measure the resistance of specimen with different lengths and analyze their relationship. The specimens were taken from the same piece of granite to reduce the deviation caused by different samples. The diameter of the specimens was 50mm, and the lengths were 25mm,

50mm, 75mm and 100mm respectively. The specimens were saturated with 10g/L sodium sulfate solution. Figure 8 shows the relationship of resistance $R$ to the ratio of length $L$ to area $S$. The square point represents the measured value, and the dotted line represents the fitting of the measured value. According to the analysis of the test circuit, the fitting curve can be expressed by the following Eq. (2):

$$R = \rho_0 \frac{L}{S} + R_e , \tag{2}$$

 

Where $\rho_0$ is the average resistivity of the specimen and $R_e$ is the additional resistance of the test line. It can be seen the resistance values of different lengths are approximately on a straight line, indicating that the specimens have similar structure. The intercept of the straight line on the Y axis is re, and its value is 7.505 kΩ. The resistance of the rocks is often in the $10^5\Omega$ level, and the additional resistance of the line is often lower than 1% of its resistance, so it can be considered that the influence of the additional resistance on the test results can be ignored.

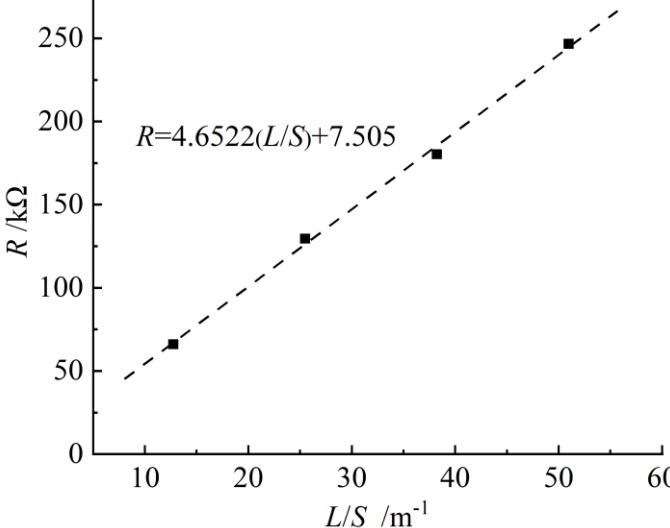


**Figure 8 Relationship of resistance to the ratio of length to area.**

**5 Preliminary experiment**

**5.1 Study of the effect of temperature and seepage pressure on resistivity**

In this section, the resistivity of fractured granite specimens was tested under different temperature and permeability pressure
conditions, the fracture penetrates along the axial direction of the specimen. In order to ensure the consistency of fracture between different specimens, rock specimens were made by the engraving method. In the test, the confining pressure was kept constant at 30 MPa, the temperature was taken as 20°C, 40°C, 60°C and 80°C, and the seepage pressure was taken as 0 MPa, 3 MPa, 6 MPa and 9 MPa. The specimens were saturated with 10 g/L of $Na_2SO_4$ solution before the test.

The variation curve of resistivity with seepage pressure is shown in Figure 9. Overall, the resistivity gradually decreases with
the increase of seepage pressure. The temperature-induced resistivity change is greater at low seepage pressure. When the seepage pressure is 0, the temperature-induced resistivity change can reach a maximum of 106.6Ω·m, and the maximum resistivity change caused by temperature is only 80.6Ω·m when the seepage pressure is 9MPa. The effect of seepage on rock resistivity is greater at lower temperatures, with a maximum resistivity change of 65.1 Ω·m caused by seepage pressure at 20°C and 39.2 Ω·m caused by seepage pressure at 80°C.



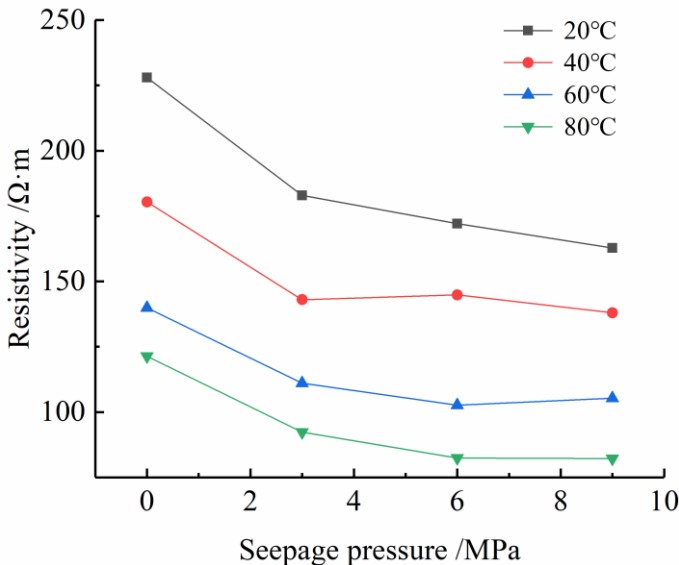


**Fig. 9 Variation curve of resistivity with seepage pressure.**

**5.2 Study on the effect of freezing and thawing on coal resistivity**

To further verify the coordinated working performance of the test system functions, triaxial tests were conducted on coal specimens under high temperature conditions and high and low temperature cycling conditions, and the resistivity was tested

during the experiments.

The high-temperature triaxial test program was: first, the specimen was sealed as described above and saturated with 10g/L of $Na_2SO_4$ solution by evacuation. Then, the specimen was loaded into triaxial cells and applied with 5 MPa confining pressure, 1 MPa seepage pressure and 50°C temperature. The triaxial test was performed after each condition reached stability for 2 hours. The loading rate of the test was 0.02mm/min, and the temperature, surrounding pressure and percolation pressure were

kept constant during the test.

The high and low temperature cycling test program was: first, the saturated specimen was maintained at 5 MPa perimeter pressure, 1 MPa percolation pressure and 50 °C for 2 h. Then the onfining pressure and seepage pressure were maintained constant and the temperature was lowered to -30 °C and maintained for 2 h. Finally, the temperature was raised to 50°C and loaded at a constant rate of 0.02 mm/min for triaxial testing.

It can be seen from Figure 10 (a) and (b) that the deviatoric stress-axial strain curves of unfrozen and frozen thawed coals are similar, showing three stages of compaction, elasticity and yield, and the axial strain values at damage are close. However, the radial deformation of coal specimen after freezing and thawing is larger than that of unfrozen specimen. The resistivity change patterns of unfrozen and frozen-thawed coals during triaxial compression are quite different. The resistivity of unfrozen coal specimen shows a decreasing trend in the initial compression stage, and then gradually increases with the raise of deviatoric

stress, but the increase amplitude gradually became smaller, and the resistivity is almost stable when it was close to the damage.





When the specimen is damaged, the resistivity first increases suddenly and then decreases rapidly. After freezing and thawing, the electrical resistivity of coal decreases in the whole compression process, and the resistivity decreases rapidly when the specimen is damaged. It can also be seen that the resistivity changes significantly when the stress-strain curve of both unfrozen and freeze-thawed specimens show large fluctuations, indicating that the resistivity changes are closely related to the strain.

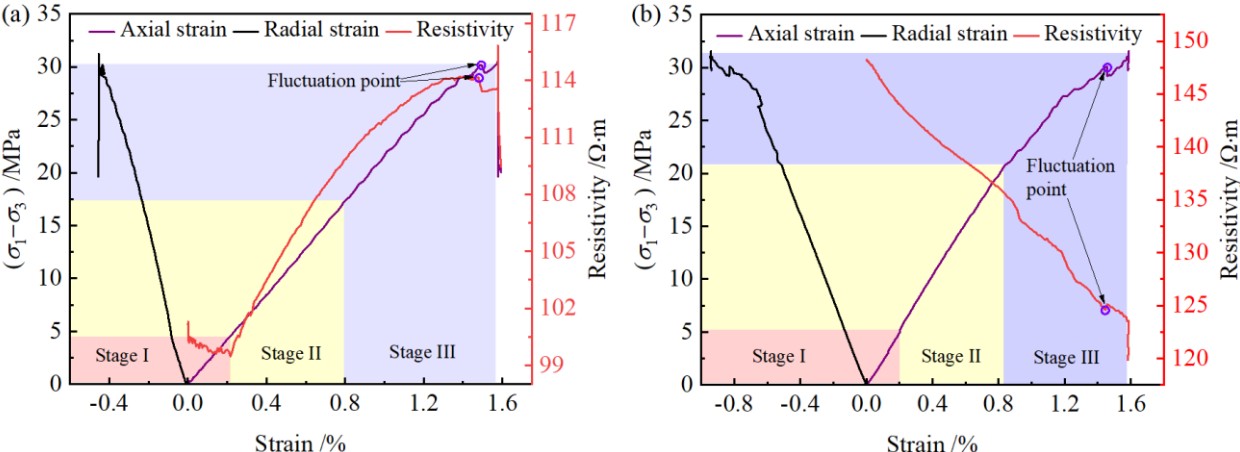


**Fig. 10 curves of stress-strain and resistivity. (a) Unfrozen coal specimen. (b) freeze-thawed coal specimen.**

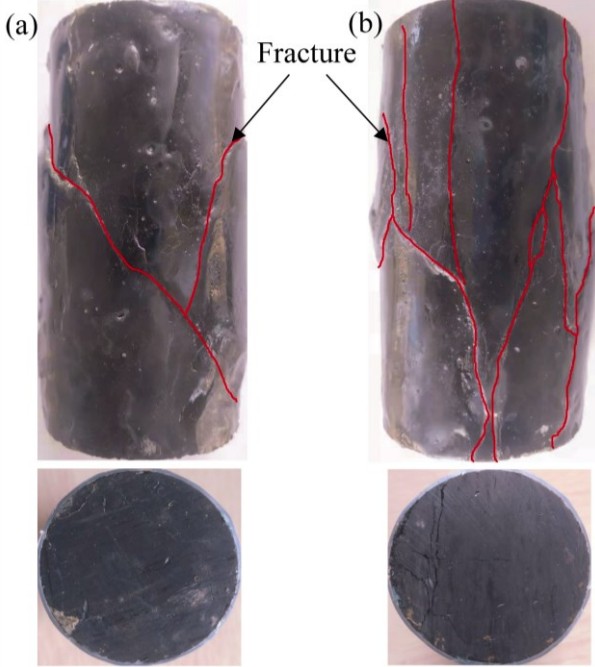

**Figure 11 fracture shape of specimens: (a) Unfrozen specimen; (b) freeze-thawed specimen**

Figure 11 shows the fracture shape of unfrozen and freeze-thaw specimens after destruction. The unfrozen and freeze-thawed
specimens show different fracture patterns. The fracture of the unfrozen specimen is crossed oblique fractures, while the

specimen after freezing and thawing has multiple axially penetrating fractures and a higher degree of specimen fragmentation. It can also be seen that there are more micro-cracks on the end face of the frozen-thawed specimen, which not only increase the content of pore water, but also may become a potential expansion channel for the specimen cracks.

## 6 Conclusion

(1) A THMC multi field rock resistivity test system is developed. The system is composed of triaxial system, chemical permeation system, temperature control system and test control system. It can simulate the thermo, hydro, mechanical and chemical environment of deep-underground rock. It provides an experimental means to study the rock resistivity characteristics and mechanical properties under different deep-underground environmental conditions.

(2) A sealing method to prevent the formation of a water film on the side of the specimen is proposed based on the
characteristics of the device. After testing, the good stability and test accuracy of the system can meet the measurement of rock resistivity under complex conditions.

(3) Preliminary experiments show that temperature and seepage pressure have significant effects on rock resistivity and have a strong coupling effect. The mechanical properties and resistivity properties of coal are greatly changed after high and low temperature effects. Therefore, it is of great significance to study the resistivity characteristics of rock under the coupling effect
of hydro-thermal-mechanical-chemical.

### Data availability

All the datasets presented in this study are available on request to the corresponding authors.

### Author contributions

Lei Song and Jianhua Yue conceived and conducted the study. Jianwei Ren, Lei Song and HaiPeng Li designed the test system.
Jianwei Ren, Qirui Wang and Junqi Fan tested the system and proposed the new sealing method. Honglei Shen and Jianhua Yue prepared the figures. Jianwei Ren wrote the main content and together with Lei Song performed the data processing. All authors contributed to the discussions and interpretation of the results.

### Competing interests

The contact author has declared that neither they nor their co-authors have any competing interests.

### Financial support

The work described in this paper was fully supported by National Natural Science Foundation of China (41974164), Special Funds for Jiangsu Science and Technology Plan (BE2022709).





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
