# Peer review of "Development of multi field rock resistivity test system for THMC"

_EGUsphere, 2022_

## Author Comment (AC1)

We would like to thank the Anonymous Referee #1 for the constructive comments concerning our article. These comments are all valuable and helpful for improving our article. According to the comments, we have tried our best to revise our manuscript. In this response letter, we have replied (in blue) to all the comments formulated by the Referee #1(in black).

**Reply on RC1**

The paper introduces the development of a multifield rock resistivity test system, which is of significance for studying the mechanical and electrical properties of rock under THMC conditions. The sealing methods for preventing the effect of water film on the testing result of resistivity were also discussed. Publication is recommended after the following minor revisions:

Our reply: We thank you for your comments and suggestions to improve the manuscript. The point by point response to all the comments and suggestions is provided in the following sections.

1. Section 5.1, the photo of engraved granite sample, as well as the size and location of the fracture are suggested to be given in the manuscript.

Our reply: Thank you for your advice. We added the photo of engraved granite sample as Figure 9 in the manuscript. The fracture in the specimen is a single fracture running from top to bottom, basically in the middle of the specimen section, and the initial average opening of the fracture is 0.1mm.

2. Section 5.1, Granite is very dense, and how to determine it is saturated?

Our reply: The saturation of granite specimens was carried out by reference to the vacuum pumping method in the *Standard for Test Methods of Engineering Rock Masses* (China standard GB/T50266-2013). First, the specimen was put into the vacuum barrel and pumped for 4h. Then, the prepared solution was injected into the vacuum barrel to make the liquid level higher than the specimen by 1cm and continued to be pumped for

more than 4h. Finally, opened the valve and let the specimen be placed under atmospheric pressure for more than 24h. In order to verify whether this method can fully saturate the specimen, we used the same method to saturate the specimen in advance and then put it for 24h, 48h and 72h for weighing. It was found that the weight has almost no change, indicating that this method is feasible to saturate granite.

3. Section 5.2, what is the cooling and heating rate?

Our reply: Because it was impossible to measure the internal temperature of the specimen during the test, the temperature of the specimen was adjusted by controlling the temperature of the confining fluid in the triaxial pressure cell. The temperature of the confining fluid in the triaxial pressure cell can be collected in real time by the temperature sensor. In the freezing and thawing test of coal specimens, the average cooling rate of confining pressure liquid was 0.2 °C/min, and the average heating rate was 0.3 °C/min.

4. 10, the changing rules of unfrozen and freeze-thawed coal samples are very different, how many samples did you test? Can the authors give some explanations for this?

Our reply: Thank you for the reminder, this is very important. As a result of your suggestion, we found the deficiencies in the current experiment. The initial purpose of the triaxial test of coal under high temperature and high-low temperature cycles was to further verify the coordinated working performance of various functions of the test system, not to study the difference in resistivity between unfrozen and freeze-thaw coal specimens. Therefore, unfrozen and freeze-thaw coal specimens were tested once respectively. However, considering that the test results showed great differences between the coal specimens before and after freeze-thawing, triaxial tests were conducted again for unfrozen and freeze-thawed coal specimens according to the same test conditions. It was found that the retest results were basically consistent with the original experimental results. The radial strain of the frozen thawed coal specimen is larger than that of the unfrozen coal specimen in the process of triaxial compression,

and the resistivity decreases gradually with the increase of deviatoric stress. The test results are shown in Figure 11.

[Figure]

**Fig. 11 curves of stress-strain and resistivity. (a), (b) Unfrozen coal specimen. (c), (b)Freeze-thawed coal specimen.**

5. Paragraph 40 and 240, "onfing" should be "confing"

Our reply: Thank you for pointing out this mistake. We have corrected it.

6. Paragraph 45, "64mpa" should be "64MPa"

Our reply: Thank you for pointing out this mistake. We have corrected it.

---

## Author Comment (AC2)

We would like to thank the Anonymous Referee #2 for the constructive comments concerning our article. These comments are all valuable and helpful for improving our article. According to the comments, we have tried our best to revise our manuscript. In this response letter, we have replied (in blue) to all the comments formulated by the Referee #2(in black).

**Reply on RC2**

This manuscript summarizes a new rock resistivity test system to undertake the relationship between rock mechanical properties and resistivity under deep underground environmental conditions. The work is interesting and can be considered for publication after going through a major review. The authors need to address the following comments:

Our reply: We thank you for your comments and suggestions to improve the manuscript. The point by point response to all the comments and suggestions is provided in the following sections.

1. On resistivity measurement and multi-field coupling test method, it is reasonable to cite three papers that are published in the International Journal of Mining Science and Technology (Spatiotemporal evolution of thermo-hydro-mechanical-chemical processes in cemented paste backfill under interfacial loading, Resistivity response of coal under hydraulic fracturing with different injection rates: A laboratory study, Creep properties and resistivity-ultrasonic-AE responses of cemented gangue backfill column under high-stress area).

Our reply: We agree with your suggestion. Citing these three papers helps to provide a more comprehensive picture of the current state of research and increases the credibility of our test methods. These three papers have been cited in appropriate sites in the manuscript.

2. Fig. 2. The design of triaxial pressure cell is conventional, please highlight innovation.

Our reply: Thank you for your advice. The new triaxial pressure cell in this manuscript was designed with reference to the original triaxial pressure cell, so the structure has similarities with the conventional triaxial pressure cell. However, the new triaxial pressure cell was designed with resistivity testing function in mind, and its structure was improved in the design. For example, the resistivity measurement is performed by the two-phase electrode method, and the current through the specimen should be accurately obtained during the test. Since the seepage tubes at both the upper and lower ends of the specimen pass through the lower flange of the triaxial pressure cell, they tend to share some of the current during measurement and cause testing errors. Therefore, a pair of PEEK sealing plugs were designed in the new triaxial pressure cell lower flange to let the seepage tube pass through its center, ensuring mutual insulation between the upper and lower seepage tubes. The enlarged view of the seal plug entity is supplemented in Figure 2.

3. Paragraph 50, can 80 °C reach the deep underground environmental temperature?

Our reply: With the development of science and technology, the depth of underground exploration has been constantly refreshed. The temperature of deep strata in oil extraction and geothermal mining has reached several hundred degrees Celsius, and the temperature exceeds 1000 °C in the study of rock properties in the mantle layer[1-2]. However, most of the current deep underground projects such as tunnels through mountains, underground laboratories and coal mining are concentrated above 2000m[3-5], and there are still many petrophysical problems to be studied in this range. The test system we designed focuses on the underground engineering problems within the depth of 2000m. According to the research, the formation temperature increases by 25 °C - 30 °C with the increase of 1000m in depth, so the temperature within 2000m often does not exceed 80 °C[6]. 80 °C meets our demand for environmental temperature.

**References**

[1] Xie H. P., Li C., He Z. Q., Li C. B., Lu Y. Q., Zhang R., Gao M. Z., Gao F.: Experimental study on rock mechanical behavior retaining the in situ geological conditions at different depths, Int. J.

Rock Mech. Min., 138, 104548, https://doi.org/10.1016/j.ijrmms.2020.104548, 2021.

[2] Aizawa Y., Ito K., Tatsumi Y.: Experimental determination of compressional wave velocities of olivine aggregate up to 1000°C at 1 GPa[J]. Tectonophysics, 339(3):473-478, https://doi.org/1016/S0040-1951(01)00133-0001, 2001.

[3] Wang Y., Jian Y. F., He Y. S., Miao Q. Q., Teng J. W., Wang Z. M., Rong L. L., Qiu L. Q., Xie C. L., Zhang Q. S., Liu X. D., Sun H. P., Yang Y. X., Yang J.: Underground laboratories and deep underground geophysical observations. Chinese Journal of Geophysics (in Chinese), 65(12): 4527-4542, https://doi.org/10.6038/cjg2022Q0404, 2022.

[4] Ling S., Ren Y., Wu X., Zhao S., Qin L.: Study on Reservoir and Water Inrush Characteristic in Nibashan Tunnel, Sichuan Province, China. In:, et al. Engineering Geology for Society and Territory - Volume 6. Springer, Cham. https://doi.org/10.1007/978-3-319-09060-3_104, 2015.

[5] Cui J. F., Wang W. J., Yuan C.: Application of stability analysis in surrounding rock control and support model of deep roadway[J]. International Journal of Oil, Gas and Coal Technology, 29(2): 180-192, https://doi.org/10.1504/IJOGCT.2022.120313, 2022.

[6] Speight J. G.: Geothermal Gradient[M]. John Wiley & Sons, Inc. 2017.

4. Paragraph 60, what is the maximum time to maintain stability, should be shown in the manuscript.

Our reply: Thank you for your advice. To meet the test requirements, the axial pressure, confining pressure, seepage pressure and temperature should be able to maintain stable for at least 48h. According to the pre-test, the above parameters can be kept stable for at least 72h. We have added this index to the manuscript.

5. Paragraph 70, the picture of the test system should be shown here.

Our reply: Thank you for your advice. We have added picture of the test system in the manuscript.

6. Paragraph 125, the temperature fluctuation of 0.5 °C has a great influence on the rock mechanics experiment.

Our reply: The accuracy of 0.5 °C refers to the temperature of the heat transfer medium in the high and low temperature circulation box, not the temperature inside the triaxial pressure cell. The temperature in the triaxial pressure cell is adjusted by the feedback of its internal temperature sensor. When the target temperature is set, the high and low temperature circulation box will heat or cool the heat transfer medium according to the measured temperature in the triaxial pressure cell, and the heat transfer medium will be driven by the pump to continuously exchange energy with the triaxial pressure cell. The temperature of the heat transfer medium changes after passing through the triaxial pressure cell, so the temperature of the heat transfer medium in the circulation box will fluctuate when it returns to the high and low temperature circulation box. The fluctuation value will decrease with the decrease of the temperature difference between the triaxial pressure cell and the heat transfer medium. According to the test, when the target temperature is reached, the temperature fluctuation inside the triaxial chamber does not exceed 0.1 °C. To avoid ambiguity, the temperature control accuracy of the triaxial pressure cell was added to the manuscript.

7. Paragraph 130, what is the temperature and pressure range of the strain gauge? How to install the flexible protective film and lead out the wire when pasting the strain gauge? How to measure and calculate the radial deformation?

Our reply: The strain gauge used in the test was a customized foil strain gauge with a resistance of 120 Ω and an applicable temperature range of -50 °C~150 °C. The pressure application range of strain gauge was not considered in the selection, but according to the previous tests, the strain gauge can still work stably under the confining pressure of 30MPa.

The flexible protective film was installed after the strain gauges were pasted, and the strain gauge wires were attached to the inside of the protective film and led from one end of the specimen. The operation steps are as follows: (1) An enameled wire of 100 mm in length and 0.1 mm in diameter is soldered on each leg of the strain gauge before the strain gauge is pasted. (2) Paste the strain gauges onto the specimen surface with

special AB glue and make the wires protrude from one end of the specimen along the specimen surface. (3) Apply rubber plastic on the surface of the specimen and the insulation cap (rubber plastic is also covered on the strain gauge and the wire). (4) Put the heat-shrinkable tube on the outside of the specimen and the insulating cap, and heat up the heat-shrinkable tube with the specimen surface by a hot air gun to make it fit completely. The sealed specimen is as shown in the S-Figure 1. Since the enameled wire is thin and completely wrapped by rubber plastic, such a sealing method will not allow the confining pressure liquid to seep into the specimen along the wire.

The radial deformation of the specimen was measured by the strain gauges perpendicular to the axial direction, and the test procedure was operated as specified in the *Standard for Test Methods of Engineering Rock Masses* (China standard GB/T50266-2013). The strain value at the strain gauge attachment site was used to represent the strain value of the whole specimen. Although the value measured by the strain gauges is the strain of the specimen perimeter, according to the relationship between the circumference and diameter of the circle, the strain of the circumference is equal to the strain of the diameter. Therefore, the collected strain value can be directly used as a measure of radial deformation.

[Figure]

Enameled wire

Heat-shrinkable tube

Strain gauge

**S-Fig. 1 Photo of sealed specimen.**

8. Paragraph 170. How does water evaporation occur in a closed triaxial pressure cell?

Our reply: The test was performed in the indoor environment, not in the triaxial pressure cell. The purpose of the test was to analyze the effect of the water film on the surface of the specimen on the resistivity test,so indoor environment evaporation was adopted to change the thickness of water film.

9. Section 5.2, there is only two sample in the triaxial test. Is the test result convincing?

Our reply: Thank you for the reminder, this is very important. As a result of your suggestion, we found the deficiencies in the current experiment. The initial purpose of the triaxial test of coal under high temperature and high-low temperature cycles was to further verify the coordinated working performance of various functions of the test system, not to study the difference in resistivity between unfrozen and freeze-thaw coal specimens. Therefore, unfrozen and freeze-thaw coal specimens were tested once respectively. However, considering that the test results showed great differences between the coal specimens before and after freeze-thawing, triaxial tests were conducted again for unfrozen and freeze-thawed coal specimens according to the same test conditions. It was found that the retest results were basically consistent with the original experimental results. The radial strain of the frozen thawed coal specimen is larger than that of the unfrozen coal specimen in the process of triaxial compression, and the resistivity decreases gradually with the increase of deviatoric stress. The test results are shown in Figure 11.

[Figure]

Fig. 11 curves of stress-strain and resistivity. (a), (b) Unfrozen coal specimen. (c), (b)Freeze-thawed coal specimen.

10. Conclusions: Please highlight the outcomes of the research.

Our reply: Thank you for your advice. We have summarized the important results obtained from the experiment and added them to the conclusion section. The modified conclusions are as follows:

(1) A THMC multi field rock resistivity test system is developed. The system is composed of triaxial system, chemical permeation system, temperature control system and test control system. It can simulate the thermo, hydro, mechanical and chemical environment of deep-underground rock. It provides an experimental means to study the rock resistivity characteristics and mechanical properties under different deep-underground environmental conditions.

(2) A sealing method to prevent the formation of a water film on the side of the specimen is proposed based on the characteristics of the device. After testing, the good stability and test accuracy of the system can meet the measurement of rock resistivity under complex conditions.

(3) Test results show that temperature and seepage pressure have significant effects on rock resistivity and have a strong coupling effect. The temperature-induced resistivity change is greater at low seepage pressure, and the effect of seepage on rock resistivity is greater at lower temperatures.

(4) The deviatoric stress-axial strain curves of unfrozen and frozen thawed coal specimen are similar, but the radial deformation of coal specimen after freezing and thawing is larger than that of unfrozen specimen. The resistivity change patterns of unfrozen and frozen-thawed coals during triaxial compression are quite different. The resistivity of unfrozen coal specimen shows a decreasing trend in the initial compression stage, and then gradually increases with the raise of deviatoric stress. After freezing and thawing, the electrical resistivity of coal decreases in the whole compression process.

11. The manuscript needs language editorial.

Our reply: We feel so sorry for the mistakes in the manuscript and inconvenience they caused in your reading. We have carefully revised the whole manuscript for grammar, spelling and punctuation, and invited native English speakers to thoroughly revise and edit the manuscript. We believe that the language level of the manuscript has improved considerably. Thank you very much for your precious comments.